

# High susceptibility of *Tetranychus merganser* (Acari: Tetranychidae), an emergent pest of the tropical crop *Carica papaya*, towards *Metarhizium anisopliae s.l.* and *Beauveria bassiana* strains

Elizabeth Alfaro-Valle[1], Aída Martínez-Hernández[1], Gabriel Otero-Colina[2] and Joel Lara-Reyna[1]

[1] Campus Campeche, Colegio de Postgraduados, Champoton, Campeche, México
[2] Campus Montecillo, Colegio de Postgraduados, Texcoco, México, México

Corresponding author
Joel Lara-Reyna, jlara@colpos.mx

## ABSTRACT

**Background**. The mite *Tetranychus merganser* is considered to be an emerging pest of various crops in tropical countries. It is one of the most detrimental pests in the papaya orchards of some regions of México. The current field control of *Tetranychus* spp. involves the extensive use of chemicals that have some degree of toxicity to humans or the environment and may cause selective resistance. The use of biological alternatives such as parasitoids and mite predators have limited effectiveness. In order to find effective but non-toxic alternatives for mite pest management, bio-products that are able to be mass produced and applied to large production areas have been sought, including the entomopathogen fungi. *B. bassiana* and *M. anisopliae* s.l. are the fungi most extensively used for the biological control of insect pests. Although they do not cause natural epizootic diseases in mites, there are reports that show that they infect *T. urticae*, and should be evaluated for use in the biological control of papaya's mite pests.
**Methods**. A *T. merganser* colony was established and the susceptibility of adult females to 30 entomopathogenic fungi strains was evaluated under laboratory conditions with an *in vitro* mass screening bioassay. Ten strains of *Metarhizium anisopliae* sensu lato (s.l.), eleven of *Beauveria bassiana*, nine of *Lecanicillium* sp. and one of *Hirsutella thompsonii* var. *sinematosa* were tested. The infectivity of adult females was evaluated calculating the percentage of mortality. To calculate the $LC_{50}$ and $LT_{50}$ of the most virulent strains, a bioassay was performed using serial concentrations ($1\times10^4$–$1\times10^8$ conidia/mL) for each strain. Strains showing ability to infect eggs laid were evaluated with a novel egg-infectivity bioassay. The internal transcribed spacer (ITS) region of the more lethal strains were sequenced.
**Results**. *T. merganser* and *T. urticae* were found in orchards of *Carica papaya* (Maradol variety and Tainung hybrid) in Campeche, México. All tested strains of *M. anisopliae* s.l. and *B. bassiana* were infectious to the adult female of *T. merganser* at a concentration of $1\times10^8$ conidia/mL. Six strains of *M. anisopliae* (Ma002, Ma003, Ma004, Ma005, Ma014 and Ma034) caused 100% mortality, and one of *B. bassiana* (Bb016) caused 95% mortality. The most virulent was Ma034, with an $LC_{50}$ of $1.73\times10^6$ conidia/mL followed by Ma005 and Ma003. Ma005 and Ma034 were the fastest strains to reach

LT$_{50}$, achieving this in less than 3.7 days. Additionally, Ma034 and Ma014 strains were infectious to more than 70% of the eggs.

**Conclusions.** *T. merganser* and *T. urticae* are present in the papaya orchards of Campeche, México. The high susceptibility of *T. merganser* adult females and eggs toward several *M. anisopliae* s.l. or *B. bassiana* strains suggests that these fungi are a viable alternative to control this emergent pest. The most virulent strain, Ma034, was also infective to eggs, and is the most promising to be tested in the field.

# INTRODUCTION

Papayas are produced in more than 60 tropical countries, with India, Brazil, Indonesia, Nigeria, México, Ethiopia, and Guatemala being the main producers (*FAO, 2020*). México is the third leading producer of papaya, representing 7.3% of the world's production (*SIAP, 2020*); it is also the main exporter. The United States and Canada are its highest importers of this crop (*SAGARPA, 2020*).

The mite complex is currently considered to be one of the most important phytosanitary problems for papaya crops in México (*Pantoja, Follett & Villanueva-Jiménez, 2002*). *Tetranychus* spp. is a complex of red spider mites, which includes the cosmopolite *T. urticae*. This is the most important pest for many crops due to the losses that it causes and its resistance to chemical pesticides (*Van Leeuwen et al., 2010*; *Adesanya et al., 2021*). In the Yucatán Península, a papaya producing and exporting region of México, the presence of the red mite has been reported by producers as an increasingly problematic pest. This problem is being addressed with pesticides that have different modes of action, such as growth inhibitors (clonfetezine and etoxazole), nerve transmission inhibitors (amitraz, spirodiclofen, avermectin), lipid sites (spiromesifen, spirotetramat), mitochondrial activity (acequinocyl, bifenazate, fenpyroximate), chitin synthesis (buprofezin), as well as ovicides and spiracle blockers such as oil paraffine and propylene glycol monolaurate. However, according to the IRAC website (https://irac-online.org/pests/tetranychus-urticae/), the resistance to avermectin, acequinocyl, bifenizate and fenpyroximate has already been reported. Additionally, some of these acaricides are toxic and producers apply them without knowledge about exactly which specie or species of mites are present. The correct identification of mites in the complex affecting a papaya crop field is necessary to select the most appropriate pest management strategies (*Abato-Zárate et al., 2014*).

*Tetranychus merganser* Boudreaux is a species that was not considered to be a significant pest because its distribution was restricted to the United States and México and its presence was occasionally reported on plants of low economic importance such as *Ligustrum vulgare*, *Solanum nigrum*, and *Solanum rostratum* (*Tuttle, Baker & Abbatiello, 1976*). However, since the last decade of the 20th century, its presence has been registered in China (*Wang & Ma, 1993*), Thailand (*Ullah & Gotoh, 2014*), and it has been related

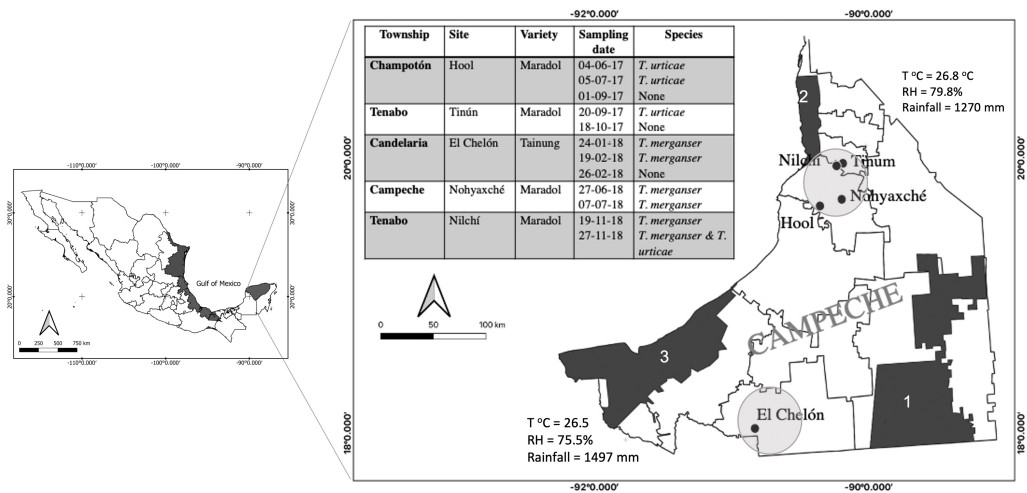

**Figure 1** ***Tetranychus* spp. identified in papaya orchards in the state of Campeche, México.** The presence of *Tetranychus* species found in the papaya orchard during the 2017–2018 season (table); sampling sites were located in the north and south of the state where the main agricultural areas were located (circles). Proximity to nature reserves and average annual climate are shown: Calakmul (1), Petenes (2) and Laguna de Términos (3); (T°C), relative humidity (RH). The presence of *T. merganser* in this work confirms its distribution along the Gulf of México (black areas).

to phytosanitary problems crops including chili peppers *Capsicum annuum* (*Estébanes-Gonzalez & Rodríguez-Navarro, 1991*), peanut *Arachis hypogaea* (*Rodríguez-Navarro, 1999*) or cactus *Opuntia ficus-indica*, in México (*Lomelí-Flores et al., 2008*; *Rodríguez-Navarro, 1999*). It has gained recent attention in the production zones of *Carica papaya* L., and it has been found in papaya orchards of several distant regions along the Gulf of México (Fig. 1). It may be the most injurious mite pest for papaya in some regions (*Valencia-Domínguez et al., 2011*, *Reyes-Pérez et al., 2013*; *Abato-Zárate et al., 2014*; *López-Bautista et al., 2016*). Thus, *T. merganser* is considered to be an emerging pest which is expanding geographically and in terms of its host range (*Lomelí-Flores et al., 2008*; *Valencia-Domínguez et al., 2011*; *Monjarás-Barrera et al., 2015*; *Monjarás-Barrera et al., 2017*).

Tetranychid mites are characterized by high fertility, arrhenotokous parthenogenesis, and short life cycles at high temperatures. Their oviposition in the abaxial side of leaves can vary with the temperature or feeding (*Dehghan et al., 2009*). At temperatures higher than 23 °C, *T. merganser* can reduce the time of all its developmental stages (egg, larva, nymph and adult). Therefore, in the tropics it is present in all seasons but its eggs lose viability at temperatures greater than 35 °C (*Hoy, 2011*; *Reyes-Pérez et al., 2013*). Its shorter life cycles, in combination with the improper handling of pesticides and their excessive use, encourages the selection of resistant populations and the reduced effectiveness of chemical acaricides (*Kim, You & Park, 1998*; *Mota-Sanchez & Wise, 2022*).

An alternative to the chemical control of pests is the use of biological agents such as entomopathogenic fungi. These biological agents are important contributors to the natural control of insect populations and have proven efficient in controlling pests in the field (*Siongers & Coosemans, 2003*; *Avery, Faull & Simmonds, 2008*). In México, several products

of microbial origin are used in a complementary manner alongside chemical pesticides to prevent population growth at levels above damaging thresholds (*Barrera et al., 2008*; *Cortez-Mondaca, 2008*; *Hernández-Velázquez & Toriello, 2008*).

Entomopathogenic fungi were named as such, considering that most of them are insect pathogens. However, since the beginning of the last century, fungi naturally infecting Acari have been reported worldwide (*Mietkiewski, Balazy & Tkaczuk, 2000*), such as species of the *Neozygites* and *Hirsutella* genera, which infect members of the Tarsonemidae, Eriophyidae, and Tetranychidae families (*Yaninek et al., 1996*; *Chandler et al., 2000*; *Vander Geest et al., 2000*; *Lopes et al., 2009*). Although these genera are specific for mites, they have characteristics that limit their mass production, commercial sale and use in the field (*Hajek, 1997*; *Poinar & Poinar Jr, 1998*). For example, some species are obligate parasites and have low conidial formation, limiting the inoculum production; and their hyphal bodies are fragile; these characteristics make it difficult to formulate. Additionally these species requires a long establishment period when they are used on pre-infected mites to inoculate a field (*Chandler et al., 2000*; *Hountondji, Sabelis & Hanna, 2010*).

*Beauveria bassiana* and *Metarhizium anisopliae* s.l. are the most commonly used fungi for pest control worldwide because they are facultative and pathogenic to a wide range of insects. Thus, it is feasible to isolate and cultivate them in the laboratory, mass-produce them on different substrates and easily apply them in the field at high spore densities. Although *B. bassiana* and *M. anisopliae* s.l. have not been found to infect mites naturally (*Wekesa, Hountondji & Dara, 2015*), and they do not cause natural epizootic diseases in mites, some isolates of *B. bassiana* and *M. anisopliae* have been shown to be infective to *T. urticae* under laboratory conditions, when inoculated in greenhouses, or in the field. This is suggestive of their potential to control this pest (*Alves et al., 2002*; *Bartkowski, Odindo & Otieno, 1988*; *Chandler, Davidson & Jacobson, 2005*; *Draganova & Simova, 2010*; *Bugeme et al., 2014*; *Bugeme et al., 2015*; *Dash et al., 2018*; *Yesilayer, 2018*). Despite the growing number of articles showing that *B. bassiana* or *M. anisopliae* s.l. are capable of infecting *Tetranychus* under controlled conditions, there are no reports to-date regarding the susceptibility of *T. merganser* to these fungi. The identification of highly-effective strains with which to infect a specific pest is a requirement for its application in biological control.

In this work, we confirm the presence of both *T. urticae* and *T. merganser* in papaya plantations of Campeche State, in the Yucatán Península, México. We report a high susceptibility of *T. merganser* to several strains of *M. anisopliae* s.l. and one *B. bassiana* also isolated in Campeche. The adult female mortality and egg infection data reported here suggest the strong potential of these entomopathogenic fungi in the biological control of this emerging pest in the field.

## MATERIALS & METHODS

### Sampling and identification of mites in papaya orchards of Campeche, México

During 2017 and 2018, five papaya orchards located in the state of Campeche, México were monitored for the presence of mites (Fig. 1). We sampled one hectare in each orchard that

**Table 1** Estimated lethal time ($LT_{50}$) of fungal isolates infecting adult females of *T. merganser*.

| Strain | $LT_{50}$(days) ± se | $FL_{lower}$ | $FL_{upper}$ | $X^2$ | m ± se |
|---|---|---|---|---|---|
| Bb005 | 4.79 ± 0.24 | 4.37 | 5.33 | 0.92 | 0.93 ± 0.25 |
| Bb014 | 5.05 ± 0.40 | 4.54 | 6.24 | 0.89 | 0.89 ± 0.24 |
| Bb015 | 5.93 ± 0.49 | 5.47 | 7.64 | 1.08 | 1.08 ± 0.57 |
| Bb016 | 3.96 ± 0.20 | 3.55 | 4.34 | 0.96 | 0.96 ± 0.25 |
| Bb019 | 6.38 ± 0.92 | 5.66 | 10.00 | 0.95 | 0.94 ± 0.48 |
| Bb021 | 6.29 ± 1.44 | 5.51 | 16.63 | 0.88 | 0.88 ± 0.38 |
| Ma002 | 3.24 ± 0.16 | 2.88 | 3.51 | 0.46 | 1.13 ± 0.49 |
| Ma003 | 4.60 ± 0.29 | 4.16 | 5.34 | 1.21 | 0.95 ± 0.27 |
| Ma004 | 3.64 ± 0.20 | 3.23 | 4.00 | 1.00 | 1.94 ± 0.35 |
| Ma005 | 3.28 ± 0.19 | 2.87 | 3.60 | 1.26 | 1.03 ± 0.39 |
| Ma006 | 4.37 ± 0.26 | 3.88 | 4.92 | 2.39 | 0.81 ± 0.17 |
| Ma007 | 3.97 ± 0.29 | 3.36 | 4.48 | 0.99 | 0.78 ± 0.16 |
| Ma008 | 4.54 ± 0.26 | 4.08 | 5.14 | 1.56 | 0.88 ± 0.20 |
| Ma009 | 5.03 ± 0.23 | 4.65 | 5.56 | 1.96 | 1.03 ± 0.37 |
| Ma014 | 3.69 ± 0.18 | 3.31 | 4.03 | 1.98 | 1.05 ± 0.37 |
| Ma034 | 3.35 ± 0.18 | 2.93 | 3.65 | 1.80 | 1.03 ± 0.38 |

**Notes.**

Bb, *Beauveria bassiana*; Ma, *Metarhizium anisopliae*; se, Standard error; FL, Fiducial limits 95%; $X^2$, Chi-square; m, slope of regression line.

was previously identified by the producers as a site with a possible mite presence. Fifteen plants were randomly selected within that area and two mature leaves were sampled from the lower stratum of each plant. The leaves were transported to the laboratory in a plastic bag inside a container. All mites present on the abaxial surface of the sampled leaves were observed with a stereoscope and collected to be identified. The specimens were incubated in lactic acid as a clearing agent, for seven days. Subsequently, the males were mounted on microscope slides with Hoyer's mounting medium (*Henderson, 2001*). The taxonomic keys of *Gerson, Smiley & Ochoa (2003)* were used to identify the families of the collected mites. The *Tuttle, Baker & Abbatiello (1976)* keys were used for identification at the genus and species level (Fig. 2A–D).

## Establishing a *T. merganser* colony

*T. merganser* were collected in papaya fields and were raised on thirty 15-day-old plants of *Phaseolus vulgaris* (Fabaceae) var. Jamapa. The plants were watered every third day. The mite colony was protected from external depredators by containing them inside cages built with a 200 μm mesh. The colony was raised at 27−28 °C, 67–77% relative humidity (RH), and a 12:12 h (light:dark) photoperiod. The density of the mite colony was maintained at approximately 200 mites per leaf. Plants severely damaged by mite infestation were replaced with new plants.

## Entomopathogenic fungal strains

Thirty strains belonging to the entomopathogenic fungi collection of the Microbial Pest Control Laboratory of the Postgraduates College (Colegio de Postgraduados, Campus

Campeche) were tested against *T. merganser*. A total of 11 strains were identified as *B. bassiana* (Bb005, and Bb014 to Bb023) and ten as *M. anisopliae* sensu lato (Ma002 to Ma009, Ma014, and Ma034) using conidiophores morphological keys. The *M. anisopliae* s.l. and *B. bassiana* strains were isolated from the soil or infected insects that were collected from several agricultural fields or semi-conserved rain forest areas from Campeche. Nine strains of *Lecanicillium* sp. isolated from *Onychiurus folsomi* (Collembola: Onychiuridae) collected in México City (19°18′58.59″N; 99°11′30.12″W) were also tested, as well as the CHE-CNRCB 377 strain of *Hirsutella thompsonii* var. *sinematosa* from the collection of the National Reference Center for Biological Control (Centro Nacional de Referencia de Control Biológico) (*SENASICA, 2016*) isolated from *Phyllocoptruta oleivora* (Acari: Eriophyidae).

The internal transcribed spacer (ITS) region of seven strains identified in this work as the more lethal for *T. merganser* (Ma002, Ma003, Ma004, Ma005, Ma014, Ma034, Bb016) were amplified by PCR using 100 ng of gDNA (*Raeder & Broda, 1985*), 40 pmol of ITS1 and ITS4 primers (*White et al., 1990*), and 30 μL of high fidelity supermix (cat 10790-020; Invitrogen, Waltham, MA). The PCR cycles consisted of 1 min at 95 °C, 55 °C and 72 °C with 2 min at 95 °C for initial denaturation and 7 min at 72 °C for the final extension. The PCR products were purified with the QIAquick PCR Purification Kit (cat. 28104; Qiagen, Hilden, Germany) and sequenced by an external service provider. The ITS sequences were compared by BLASTn analysis with NCBI (nr/nt) and UNITE 8.3 (*Nilsson et al., 2019*) databases. The *M. anisopliae* s.l. ITS sequences were aligned with ITS sequences of reference strains from UNITE/NCBI databases with Clustal Omega (1.2.4) and a guide tree was constructed. A phylogenetic tree was constructed with MEGA X using the UPGMA and Neighbor-Joining method with 1,000 replicates.

The strains of *Beauveria*, *Metarhizium*, and *Lecanicillium* were grown in potato dextrose agar and incubated at 28 °C until conidiogenesis occurred. *H. thompsonii* was grown in Sabouraud dextrose agar with yeast extract and transferred to medium H (*Cabrera, Vega & Ayra, 2006*) for sporulation. The conidia were recovered from sporulated cultures with 0.01% triton in sterile distilled water by scraping the surface. The suspension was homogenized and the conidia were counted in a Neubauer chamber to adjust the concentration. The percentage of viable conidia was also determined. Isolates with more than 90% germination were used in bioassays.

## Screening of fungal pathogenic strains on adult *T. merganser* females

To evaluate whether any *B. bassiana*, *M. anisopliae* s.l., or *Lecanicillium* sp. strains from our collection or the *H. thompsonii* strain from SENASICA were pathogenic to *T. merganser*, we performed a mass screening bioassay evaluating mortality on adult females (Fig. 2A). For this infectivity bioassay, papaya leaf disks of three cm in diameter were disinfected under aseptic conditions in a solution of sterile distilled water with sodium hypochlorite (0.06%) and 10 μL of liquid soap, and continually stirred for 1 min. Leaf disks were rinsed three times with sterile distilled water and subsequently allowed to dry on a sterile paper towel. Each leaf disk was immersed in 2 mL of $1 \times 10^8$ conidia/mL of each strain for 30 s, before allowing it to dry on a sterile paper towel. This conidia concentration

was previously selected by testing three strains at $1 \times 10^4$, $10^6$, and $10^8$ conidia/mL. We observed a mortality of more than 50% only at the highest concentration. All strains of the *Lecanicillium* sp. showed very low sporulation levels, thus we used $1 \times 10^7$ conidia/mL for bioassays. The viability of the conidia of all tested strains was evaluated before each experiment. The inoculated disks were placed upside down on an Oasis® floral sponge base of $3.6 \times 3.3 \times 0.5$ cm inside a five cm diameter Petri dish containing 3 mL of water. Twenty adult *T. merganser* females were placed onto each disk using a fine brush. In the control group the disks were immersed in sterile distilled water with 0.01% triton. The Petri dishes were kept closed and maintained at 28 °C with 50–55% RH and a 12:12 h (light:dark) photoperiod. On the third day, the mites were transferred onto a new leaf disk without inoculum. The mortality of the mites was quantified daily from the third to the sixth day post-infection. The mortality caused by mycosis was verified by the microscopic observation of specific fungal sporulation on the mummified mites (Fig. 2E). Three independent assays comparing all strains were performed. The average percentage and the standard deviation of adult mortality caused by each strain were calculated and then plotted with Sigma Plot 14.0. Afterwards, the percentage of mortality was arcsine-transformed to normalize the data before one-way analysis of variance (ANOVA) was performed. Differences between the means were compared by Tukey tests, and $p = 0.05$ was considered as significant. To estimate the lethal time to 50% mortality ($LT_{50}$) with the screening assay data, the accumulated mortality recorded at the concentration of $1 \times 10^8$ conidia/mL for each strain were corrected for natural mortality (*Abbott, 1925*) and used to estimate $LT_{50}$ using the Probit function (*Finney, 1971*) with a significance level of 0.05%. The statistical analyses of all bioassays in methods section were carried out using *Minitab 14 Statistical Software (2003)*.

## $LC_{50}$ & $LT_{50}$ of the most virulent strains

The seven more virulent strains (causing 90–100% mortality of *T. merganser* in less time) were selected to perform a bioassay similar to the one described above. Each strain was evaluated in serial concentrations from $1 \times 10^4$ to $1 \times 10^8$ conidia/mL, from the third to the seventh day. In this range, the recorded mortality was between 0 and 100%. For each strain, at least six independent experiments were performed and the lethal concentration to 50% mortality ($LC_{50}$) was estimated and the $LT_{50}$ was recalculated using the Probit analysis with a significance level of 0.05%.

## Infectivity on eggs

The strains that showed the ability to infect eggs laid by the females during the screening assay (Fig. 2F) were evaluated with a novel infectivity bioassay. Sixty adult females were placed on a disinfected papaya leaf fragment ($8 \times 5$ cm) for 24 h to lay eggs. The females were removed by washing the leaves with 0.05% triton with strong agitation for 1 min. The leaves were allowed to dry in a laminar flow cabinet. Twenty eggs were carefully recovered with a fine brush and placed onto 1 cm$^2$ of sterilized screen-printing mesh (130 µm). The mesh was previously placed onto 1 cm$^2$ of a sterilized cotton pad and both were placed onto a sterilized glass slide. The eggs were separated with the brush, while avoiding contact

between the eggs (Fig. 2G). A volume of 1.5 μL of a $1 \times 10^8$ conidia/mL suspension was added onto the glass slide, which was sufficient to cover all eggs, allowing the excess liquid to be absorbed by the cotton pad. Sterile distilled water with 0.01% triton was added to eggs in the control group. Each slide was placed onto a cotton pad moistened with sterile distilled water to promote fungal growth, inside a 15 mm Petri dish. The Petri dishes were kept at 28 °C, 50–55% RH, and a 12:12 h (light:dark) photoperiod. Infectivity was recorded on the fifth day based on the presence of mycelia on the eggs (Fig. 2H) and by verifying the characteristics of the spores of each fungus. The fungi were stained with lactophenol cotton blue at 0.5%. Three independent experiments were performed. The average percentage and the standard deviation of infectivity on eggs were calculated, and plotted (Sigma Plot 14.0). The percentage of infectivity was arcsine-transformed and a one-way analysis of variance (ANOVA) was performed. Differences between the means were compared by Tukey tests; $p = 0.05$ was considered as significant.

## RESULTS

### Identification of *T. merganser* and *T. urticae* in papaya orchards of Campeche, México

Five papaya orchards located in two distant sites in the Campeche State were sampled repeatedly over two years, reaching a total of twelve samples (Fig. 1). The presence of two *Tetranychus* species was recorded: *T. merganser* and *T. urticae*. On some sampling dates, we did not find any mites, possibly because the producers use acaricides to control the pest. *T merganser* was found in high density in three of the five sampled orchards growing on two different papaya varieties: Maradol and a Tainung hybrid. *T. urticae* was observed in less density than *T. merganser*, when it was present. The characteristics confirming the identification of species from the *Tetranychus* genus are the distance between the pair of duplex setae, the presence or absence of tenant hair, and the morphology of the aedeagus knob. The aedeagus knob of *T. urticae* has an anterior projection acute and a rounded shape in posterior projection, their females are greenish (Figs. 2B, 2D). The aedeagus of *T. merganser* resembles the head and neck of a goose (Fig. 2C); the pretarsus do not have tenant hair, and each empodium ends in a tuft of three pairs of hair. The specimens that were identified as *T. merganser* had the unique coloration of its species, namely bright red females and pale yellowish males (Fig. 2A).

Given that *T. merganser* has been previously reported in *C. papaya* orchards of Yucatán and Veracruz (Fig. 1), other states located along the Gulf of México (*Valencia-Domínguez et al., 2011*; *Abato-Zárate et al., 2014*; *Monjarás-Barrera et al., 2017*), we looked for biological control for this emerging pest to avoid the use of chemical acaricides and resistance selection.

### Screening of fungal pathogenic strains on adult *T. merganser* females

The *H. thompsonii* strain that was tested showed no infectivity in *T. merganser* adult females at $1 \times 10^8$ conidia/mL, even on the sixth day post-inoculation. However, many strains in our collection were infectious towards adult *T. merganser* females, indicating a considerably different mortality effect between species (Fig. 3). The *M. anisopliae* s.l. strains caused the
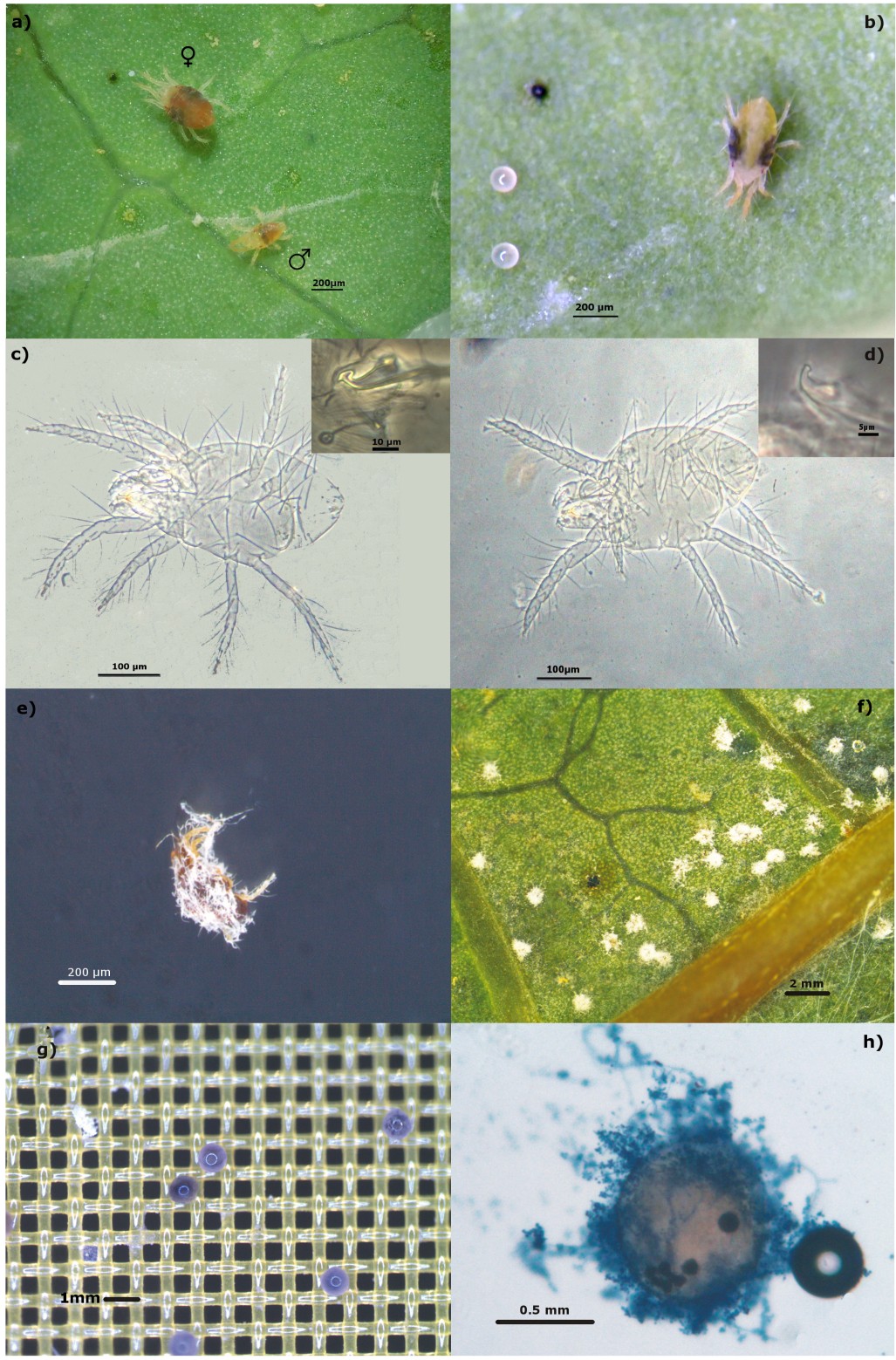

**Figure 2** **Identification of *Tetranychus merganser* and bioassays.** (A) *T. merganser* showing bright red females and pale yellowish males while *T. urticae* (B) their females (continued on next page…)

**Figure 2 (…continued)**
are greenish, these females were used in the mass screening for bioinfectivity; (C) *T. merganser* (male) showing aedeagus knob resembles the head and neck of a goose (box); (D) *T. urticae* (male) showing the aedeagus knob of has an anterior projection acute and a rounded shape in posterior projection (box); (E) *T. urticae* mummified for fungus; (F) eggs infected in screening assay, (G) eggs on mesh (130 μm) for the infectivity bioassay, and (H) egg infected and stained with lactophenol cotton-blue.

highest *T. merganser* mortality levels in comparison to the other fungal species (Fig. 3A). All these strains resulted in significant differences in mortality, compared to the control ($F = 74.85$, $p < 0.0001$) (Data S1). Five *M. anisopliae* s.l. strains (Ma002, Ma004, Ma005, Ma014 and Ma034) caused mortality to 100% of mites, Ma003 to 91%, and the other strains (Ma006, Ma007, Ma008 and Ma009) caused mortality from 75% to 85%.

In the case of *B. bassiana* (Fig. 3B), all strains were infectious to adult females, but only seven of the eleven strains showed mortality with statistically significant differences compared to the control ($F = 17.55$, $p < 0.0001$) (Data S1). Three strains (Bb005, Bb014, Bb016) showed the highest mortality levels, reaching 95% in the case of Bb016. The Bb015, Bb019 and Bb021 strains showed a mortality of only 40–43%.

Very low sporulation levels were observed in the strains of the *Lecanicillium* sp., thus we were not able to obtain a highly concentrated suspension, but performed the test at $1 \times 10^7$ conidia/mL. At this concentration, no strain of *Lecanicillium* sp. showed mortality with a statistically significant difference from the control ($F = 2.9$, $p < 0.023$) (Fig. 2C and Data S1), although all strains showed some degree of infectivity (mycosis on the mummified mite).

To compare the fungal strains' speed to kill *T. merganser* adult females, we calculated the time it took to reach 50% of mortality ($LT_{50}$) at $1 \times 10^8$ conidia/mL with strains that showed significant differences in mortality, compared to the control (Table 1). In this preliminary assay, the five *M. anisopliae* s.l. strains that caused 100% mortality to *T. merganser* also showed the fastest mortality times (Ma002, Ma005, Ma034, Ma004, and Ma014). Of the *B. bassiana* strains, Bb016 exhibited the shortest $LT_{50}$, followed by Bb005 and Bb014. These are also the three *B. bassiana* strains that caused the highest mortality to *T. merganser*. The chi-square, fiducial limits (FL) and slope (m) from Probit analysis are shown in Table 1.

## LC$_{50}$ & LT$_{50}$ of the most virulent strains

We selected the six strains of *M. anisopliae* s.l. (Ma002, Ma003, Ma004, Ma005, Ma014 and Ma034) and the one of *B. bassiana* (Bb016) that caused 90%–100% mortality to *T. merganser* and showed the lower $LT_{50}$ in the mass screening assay to evaluate their virulence with serial doses for seven days and with more replicates (Data S1). The doses required to cause mortality to 50% of a population ($LC_{50}$) were determined and the $LT_{50}$ was recalculated. The most virulent strains were Ma034, Ma005, Ma003, Bb016 and Ma002, with $LC_{50}$ values around $10^6$ conidia/mL ($1.7-3.2 \times 10^6$, respectively) (Table 2); which also have the lowest $LT_{50}$. The Ma004 and Ma014 strains had $LC_{50}$ values above $10^7$ conidia/mL and the highest $LT_{50}$. The chi-square, fiducial limits (FL) and slope (m) from Probit analysis for $LC_{50}$ are shown in Table 2. Taken together, the $LT_{50}$, and $LC_{50}$ data

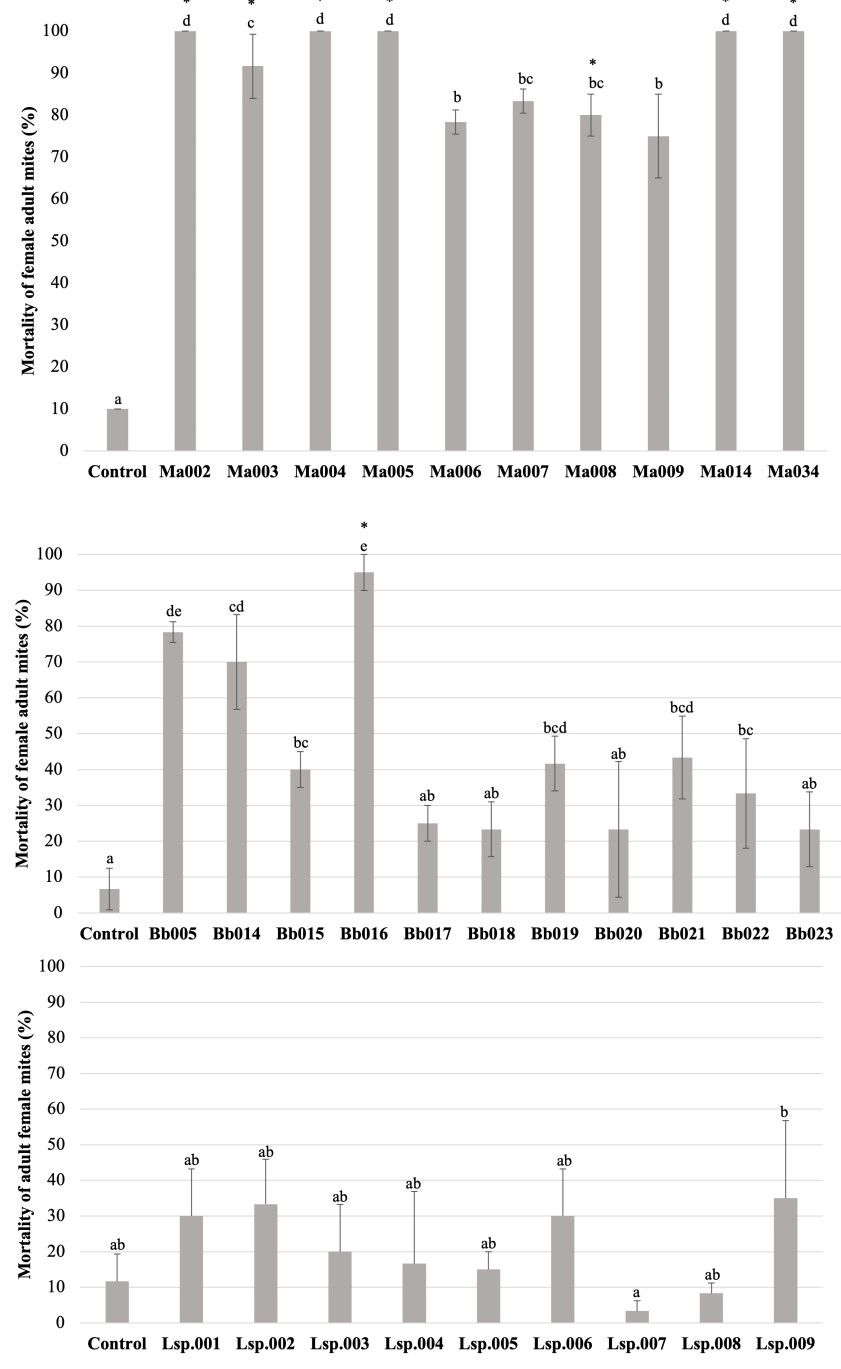

**Figure 3** Percentage mortality of *T. merganser* by different strains of entomopathogenic fungi (1 × 10⁸ conidia/ml). (A) *M. anisopliae* s.l., (B) *B. bassiana*, and (C) *Lecanicillium* sp. (1 × 107 conidia/mL). Asterisk: strains that were infective towards eggs. Columns with different letters are significantly different among strains based on Tukey's test ($p < 0.05$) after one-way ANOVA analysis. Bars: standard error.

**Table 2** Median lethal concentration (LC50) of more lethal strains for *T. merganser* adult females.

| Strain | $LC_{50}$ conidia/ml | $FL_{lower}$ | $FL_{upper}$ | X2 | m ± se | LT50 ± se |
|---|---|---|---|---|---|---|
| Ma034 | 1.73E+06 | 7.94E+05 | 3.85E+06 | 2.38 | 1.04 ± 0.17 | 3.70 ± 0.20 |
| Ma005 | 1.97E+06 | 9.13E+05 | 4.34E+06 | 2.22 | 1.03 ± 0.17 | 3.68 ± 0.19 |
| Ma003 | 2.32E+06 | 1.03E+06 | 5.56E+06 | 2.29 | 1.04 ± 0.17 | 3.94 ± 0.24 |
| Bb016 | 2.62E+06 | 1.32E+06 | 5.35E+06 | 1.11 | 1.39 ± 0.26 | 3.91 ± 0.22 |
| Ma002 | 3.21E+06 | 1.53E+06 | 6.96E+06 | 1.88 | 1.20 ± 0.21 | 3.93 ± 0.21 |
| Ma004 | 1.26E+07 | 5.13E+06 | 4.49E+07 | 0.78 | 0.87 ± 0.17 | 4.72 ± 0.37 |
| Ma014 | 5.27E+07 | 1.54E+07 | 9.13E+08 | 1.20 | 0.70 ± 0.16 | 6.11 ± 0.41 |

Notes.
$X^2$, Chi-square; m, slope of regression line ± standard error; FL, Fiducial limits 95%.

**Table 3** Highly virulent strains to *T. merganser* selected from the screening and BLASTN data.

| Strain | Source | Geolocation | Molecular identification | Score (Bits) | *E* value | Genbank accession | Clade |
|---|---|---|---|---|---|---|---|
| Bb016 | Soil | 18°35′35.2″N 90°01′58.2″W | *Beauveria bassiana* | 480 | 2e−134 | OM490017 | |
| Ma002[*] | Chrysomelinae[a] | unknown | *Metarhizium anisopliae* | 904 | 0.0 | OM490011 | PARB |
| Ma003[*] | Spodoptera larvae[a] | unknown | *Metarhizium anisopliae* | 713 | 0.0 | OM490012 | PARB |
| Ma004 | Cercopidae | 18°32′50.5″N 89°54′28.8″W | *Metarhizium robertsii* | 558 | 1e−157 | OM490013 | PARB |
| Ma005 | Soil | 18°33′06.4″N 89°54′20.5″W | *Metarhizium robertsii* | 671 | 0.0 | OM490014 | PARB |
| Ma014[*] | Coccineliidae[a] | unknown | *Metarhizium anisopliae* | 743 | 0.0 | OM490015 | PARB |
| Ma034[*] | Cercopidae[a] | 18°32′46.8″N 89°54′45.6″W | *Metarhizium robertsii* | 564 | 3e−159 | OM490016 | PARB |

Notes.
[*]Collected by Comité Estatal de Sanidad Vegetal Campeche (Cesavecam).
E, expect value.
[a]Unidentified species.

allowed us to identify Ma034 and Ma005 as the most aggressive strain towards *T. merganser* adult females followed by Ma003, Bb016 and Ma002.

The ITS of these seven strains were sequenced and we confirmed the identification of Bb016 as *B. bassiana* (Table 3). According with BLASTn hits, ITS from three strains of *M. anisopliae* s.l. (Ma002, Ma003 and Ma014) showed the highest similarity with *M. anisopliae* and three with *M. robertsii* (Ma004, Ma005 and Ma034) (Table 3). The phylogenetic analysis confirmed that these six strains belong to the so-called PARB clade of *M. anisopliae* (Table 3, Fig. S1). Altough the ITS information is not enough to distinguish between the species of PARB clade (*Bischoff, Rehner & Humber, 2009*; *Rehner & Kepler, 2017*); the ITS alignment showed that Ma004, Ma005 and Ma034 have a different number of adenines in a short stretch of this nucleotide, comparing with Ma002, Ma003 and Ma014 (Fig. S2). That difference is also present in other *M. robertsii* ITS sequences from the Genbank or UNITE databases, including sequences from the reference strains ARSEF 2575 and ARSEF 23, compared to *M. anisopliae* sequences from ARSEF-7487, JEF-290, BRIP 53293, BRIP 53284.

## Infectivity on eggs

We recorded that seven *Metarhizium anisopliae* s.l. strains (Ma002, Ma003, Ma004, Ma005, Ma008, Ma014 and Ma034) and two *B. bassiana* strains (Bb016 and Bb017) were infectious

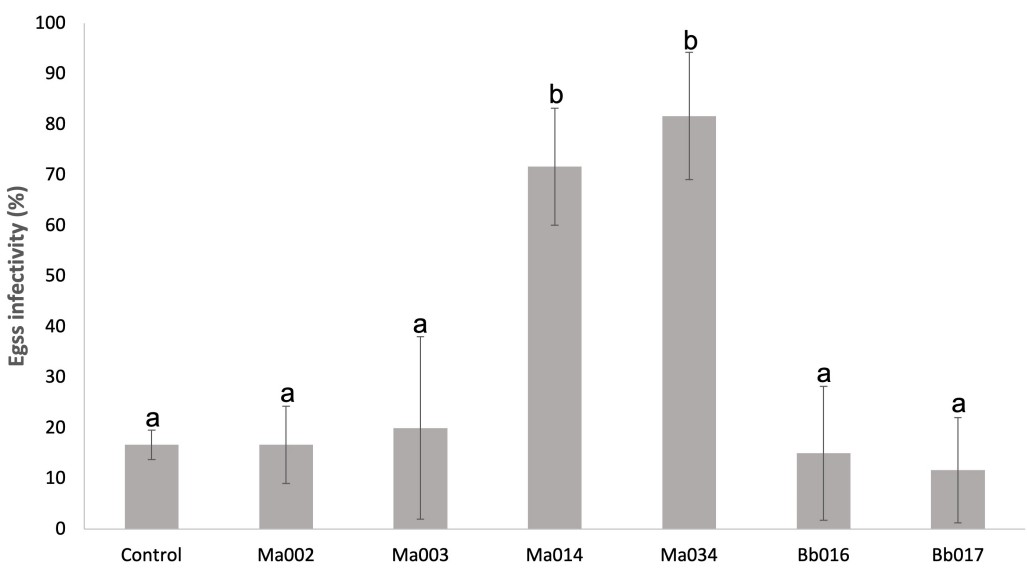

**Figure 4** **Infectivity on eggs of *T. merganser* of six strains of entomopathogenic fungi.** Columns with different letters are significantly different among strains based on Tukey's test ($p < 0.05$) after one-way ANOVA analysis. Bars: standard error.

to *T. merganser* eggs laid as an additional observation during the screening assay (Figs. 2F and 3). Most of these strains also showed a high mortality to the adult females. Thus, we developed an *in vitro* egg infectivity assay (Fig. 2G).

All of the tested strains showed mycosis on the eggs, although four strains (Ma002, Ma003, Bb016 and Bb017) did not show statistically significant differences compared to the natural mortality in the control group ($F = 14.83$, $p < 0.0001$) (Fig. 4 and Data S3 ). However, the Ma014 and Ma034 strains showed a high infection level, reaching 71.7 and 81.7%, respectively (Data S3 ). The data of the Ma004, Ma005 and Ma008 strains were excluded from the analysis due to their high variability. This analysis points to Ma034 as the most promising strain since it is the most virulent to adult females with 100% mortality, has the lowest $LC_{50}$ and low $LT_{50}$ value, and is highly infectious to eggs.

## DISCUSSION

We confirmed the presence of the emergent pest *T. merganser* in papaya orchards in a new region of southeastern México and proposed biological alternatives using entomopathogenic fungi for its control. Through taxonomical identification, we found two *Tetranychus* species in the orchards sampled in this study: *T. urticae* and *T. merganser*. The presence of *T. urticae* was expected, since it is a cosmopolite pest for many crops and it was previously reported in papaya fields (*Abato-Zárate et al., 2014*; *Mena et al., 2017*). Although papaya producers have reported that the red spider mite is the predominant pest of this crop in Campeche, the presence and identity of these two *Tetranychus* species had not been confirmed in this region. However, *T. merganser* was previously reported in commercial papaya crops of other regions along the Gulf of México (*Valencia-Domínguez et al., 2011*;

*Abato-Zárate et al., 2014*; *Monjarás-Barrera et al., 2017*). The presence of high densities of *T. merganser* in papaya orchards in the north and southwest of Campeche suggests that the mites could be more widespread than currently recorded. They may be found throughout the entire Yucatán Península and other surrounding regions. Furthermore, its persistence measured over two years confirms its status as an emerging pest and highlights the importance of seeking strategies to control it in the field. The presence of this emergent pest in other tropical crops must be studied to determine its impact there. To find a biological alternative for the control of *T. merganser* and to reduce the use of chemical acaricides, responding to the requests of local papaya producers, we evaluated thirty entomopathogenic fungal strains from our collection: *M. anisopliae* s.l., *B. bassiana,* and *Lecanicillium* sp. We compared these with *H. thompsonii* var. *sinematosa* isolated from a mite. The CHE-CNRCB 377 strain did not show any infectivity towards *T. merganser*, despite *H. thompsonii* var. *sinematosa* being previously described as a potential agent to control mites (*Quesada-Sojo & Rivera-Méndez, 2016*), including Tetranychidae (*Rosas, 2003*); this may be due to CHE-CNRCB 377 being isolated from a mite of a different family (Acari: Eriophyidae) (*Rosas-Acevedo & Sampedro-Rosas, 1990*; *Cabrera, Ayra & Martínez, 2007*).

Although *Lecanicillium lecanii* was reported as a fungus with a natural incidence in *Tetranychus* (*Chandler et al., 2000*), none of our *Lecanicillium* sp. strains showed the potential to be applied in field against *T. merganser*. Our *Lecanicillium* sp. strains caused infection to *T. merganser* with low mortality, in agreement with findings of *Chandler, Davidson & Jacobson (2005)*. In addition, they have low sporulation levels and the predicted $LT_{50}$ is very long. All of these characteristics exclude them from mass production and use in the field.

However, despite not being isolated from mites under natural conditions (*Wekesa, Hountondji & Dara, 2015*), all of the *Metarhizum anisopliae* s.l. strains and many of the *B. bassiana* strains that were tested showed significant lethality to adult females. This indicates that *T. merganser* is susceptible to these entomopathogenic fungi. Previous reports have shown the infectivity of these two species against other members of the Tetranychidae family, mainly *T. urticae* (*Draganova & Simova, 2010*; *Seiedy et al., 2010*; *Wekesa, Hountondji & Dara, 2015*; *Habashy et al., 2016*; *Shin et al., 2017*). This infectivity is feasible due to the artificial interaction between host-pathogen that may be established between this fungal group (Hyphomycetes) and Acari at high conidial concentrations. The surface structure and chemical composition of the host cuticula likely determines the success of the interaction (*Pedrini, Crespo & Juárez, 2007*). Since the mites have similar protection barriers as insects (*Chandler et al., 2000*), the entomopathogenic fungi could infect them through similar mechanisms.

Our data show that all the *M. anisopliae* s.l. tested strains and six of *B. bassiana* were infectious to adult females of *T. merganser*. Additionally, *T. merganser* at this stage was highly susceptible (95–100% of mortality and short $LT_{50}$ values) to six strains of *M. anisopliae* s.l. and one of *B. bassiana*. Variations in the effectiveness between strains of a same fungal specie is a common phenomenon, possibly due to the high genetic and metabolic diversity between strains (*Wang & St. Leger, 2005*; *Xiao et al., 2012*; *Pattemore et al., 2014*;
*Wang et al., 2016*; *Zhang et al., 2021*). In comparison with other reports, the performance of our assay after adding the fungus by immersing the leaf-disks was comparable to a spray application (*Wekesa et al., 2005*; *Zhang, Shi & Feng, 2014*). The strains tested here were more lethal and virulent (lower $LC_{50}$) against *T. merganser* than other strains of both fungi tested against *T. urticae* (*Bugeme et al., 2014*). Other studies show similar levels of virulence for *T. urticae* than our strains for *T. merganser* (*Tamai, Alves & Neves, 1999*; *Alves et al., 2002*; *Wekesa et al., 2005*; *Chandler, Davidson & Jacobson, 2005*; *Habashy et al., 2016*). We found that more strains of *M. anisopliae* s.l. than of *B. bassiana* were highly infective to *T. merganser*, however, other studies have found *B. bassiana* strains more virulent to *T. urticae* than all *Metarhizium* strains reported (*Seiedy et al., 2010*). Thus, strains of both fungi species seem to be highly infective against *Tetranychus* species.

Finally, since egg control is important to decrease or avoid future outbreaks in the field, we evaluated the infectivity of our strains on eggs. Entomopathogenic fungi have been reported to have the ability to infect insects in all stages, including eggs (*Hajek & St. Leger, 1994*), which is also the case for mites. The infection process in the eggs is similar to that which occurs in the body of arthropods, without the development of blastospores inside the egg as occurs in the hemocele of insects (*Zhang, Shi & Feng, 2014*). Several studies have evaluated the susceptibility of *T. urticae* eggs to different entomopathogenic fungal species. It has been found that the egg response varies from extremely lethal to non-infectious, depending on the strain (*Shi & Feng, 2004*; *Bugeme et al., 2014*; *Zhang, Shi & Feng, 2014*; *Dogan et al., 2017*). Although nine of our strains showed infectivity towards *T. merganser* eggs, only two *Metarhizium* strains (Ma014 and Ma034) showed consistently high infectivity (above 70%) towards eggs. This ovicidal activity is relevant, because the pest cannot regenerate if a high percentage of eggs die in the population. The use of strains which combine a high virulence to adult females with a high egg mortality, would allow a more effective control of *T. merganser* in crops.

Our data show that five strains (Ma034, Ma005, Ma003, Bb016 and Ma002) are the most virulent to adult females (lowest $LC_{50}$). Ma034 and Ma005 are the most aggressive (100% mortality and lowest $LC_{50}$ & $LT_{50}$). Additionally, Ma034 showed consistently high infectivity towards eggs, which allowed us to select it as the most promising for field testing. Although Bb016 was among the most lethal strains with a low LC50 value, the low infectivity on eggs is an important parameter for its efficacy in the field, so its use could be combined with other egg-infecting strains.

The most aggressive *Metarhizium* strains described in this work belong to the PARB clade. *M. anisopliae* s.l. is a complex of 12 species, among them the clade informally called PARB that includes *M. pingshaense*, *M. anisopliae*, *M. robertsii* and *M. brunneum*, only distinguishable with a multilocus phylogenetic analysis (*Bischoff, Rehner & Humber, 2009*; *Rehner & Kepler, 2017*). These type species represent the core of the *M. anisopliae* complex, are the most frequently isolated from insects and soils, have a wide host range and a global distribution (*Bischoff, Rehner & Humber, 2009*; *Rehner & Kepler, 2017*). It is important to note that the strains described here as infective to *T. merganser* are native to the region where this emergent pest has been reported, reducing the risks to other non-target arthropods.

## CONCLUSIONS

*T. merganser* is an emerging pest present in the papaya crops of Campeche, México, in addition to *T. urticae*. Adult females and eggs of *T. merganser* are highly susceptible to some strains of the entomopathogenic fungi *M. anisopliae* (PARB clade) and *B. bassiana*. Five strains identified here are as the most infective and virulent against *T. merganser* have good potential to be used in integrated management programs to control this pest. The most aggressive strain, Ma034, also infective for eggs, is the most promising to be tested in papaya fields. The beneficial effects of these entomopathogenic fungi on the economic impact of this pest or on the use of toxic chemicals remains to be addressed.

## ACKNOWLEDGEMENTS

We thank Pech-Chuc for his technical assistance with the PCR analysis of fungal strains. The papaya producers, F. J. Medina and N. Tress Marini, for the opportunity to sample mites in their orchards, and Ph. D. Victor Hugo Quej-Chi for help with Fig. 1.

### Funding

This work was supported by the CONACYT fellowship 619249 awarded to Elizabeth Alfaro-Valle. The funders had no role in study design, data collection and analysis, decision to publish, or preparation of the manuscript.

### Grant Disclosures

The following grant information was disclosed by the authors:
CONACYT: 619249.

### Competing Interests

The authors declare there are no competing interests.

### Author Contributions

- Elizabeth Alfaro-Valle conceived and designed the experiments, performed the experiments, analyzed the data, prepared figures and/or tables, and approved the final draft.
- Aída Martínez-Hernández conceived and designed the experiments, analyzed the data, prepared figures and/or tables, authored or reviewed drafts of the article, sequence analysis and molecular identification of strains, and approved the final draft.
- Gabriel Otero-Colina conceived and designed the experiments, authored or reviewed drafts of the article, and approved the final draft.
- Joel Lara-Reyna conceived and designed the experiments, performed the experiments, analyzed the data, prepared figures and/or tables, authored or reviewed drafts of the article, and approved the final draft.

## DNA Deposition

The following information was supplied regarding the deposition of DNA sequences:

The sequences are available at GenBank: OM490011 to OM490017.

## Data Availability

The raw data is available in the Supplementary Files.

## Supplemental Information

Supplemental information for this article can be found online at http://dx.doi.org/10.7717/peerj.14064#supplemental-information.

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
