# Peer review of "High susceptibility of Tetranychus merganser (Acari: Tetranychidae), an emergent pest of the tropical crop Carica papaya, towards Metarhizium anisopliae s.l. and Beauveria bassiana strains"

_PeerJ, doi:10.7717/peerj.14064_

## Round 0.1 · original submission · Major Revisions

Please note that the reviewers recommended that the manuscript needs a major revision for improved clarity, conciseness and readability.

Reviewer 2 has suggested that you cite specific references. You are welcome to add it/them if you believe they are relevant. However, you are not required to include these citations, and if you do not include them, this will not influence my decision.

Reviewer 1 ·

Basic reporting

Major points
1. At the end of your discussion you specify that you suggest two M. robertsii strains as being the most virulent and are the two most promising strains to use – can you highlight this more in your title and in the conclusion of your abstract
a. I see you do mention it in the conclusion of the abstract – but I think you should make it even more clearer these two strains are the most promising.
b. And then specify that strain MA034 was the most virulent of all of them. Or just in general clarify that there are two promising strains to use in field and one of them is better than the other.
c. The title includes the susceptibility against the other strains, however you really only consider one of the species as most important to study.
2. GenBank accession number given but is not made available yet, please attach the files as a supplement.
a. There is no data availability statement with the GenBank accession number in the manuscript.
3. Please include a section with information and description of each of the supplemental files.
4. Methods portion of abstract does not contain strain information or mention Metarhizium robertsii
a. Identify in methods that the ITS identification separated out the species and you cultivated M. robertsii strains when you thought they were M. anisopliae or make it very clear the strains are different species
i. I know you are using the term sensu lato, but I think it muddles the interpretation of the results by not keeping the species completely separate in the methods and results
b. It is confusing to go back to methods and the results of the ITS information, leading us to believe you used one species when it turns out it was another
c. Highlight in figures which strains are which species as well and keep this distinction consistent throughout
5. Mention Tetranychus urticae in abstract as well as T. merganser
6. Can you include a little background information in the intro section on how these specific fungi could/should control the mites biologically?
a. Expand in discussion section for these specific strains
b. Are there currently any fungal strains being used against these mites in the field – I know you state that some fungal isolates have been found infective towards the mites but it is not clear if anyone is using any kind of fungal control on these mites
7. Can you include more reproductive biology of the mite species in the intro?
a. How many eggs they lay, where on the plants, seasonality, etc.
8. What are the current pesticides in use specifically at these locations? (intro)
9. Lines 356-358 should not be tacked on at the end as an afterthought, these should be highlighted in the intro section as a positive statement as to the reason you are using these strains

Minor points
10. Line 20 – [such] as papaya
11. Line 21 - implies should be employs
12. Line 23 – “The biological practices…” what do you mean by biological practices? Please give more detail about this.
13. Line 28 – “e”stablished
14. Line 55 – remove “(third place in the world)” and just state that Mexico is the 3rd leading producer of papayas directly
15. Line 64 – but not as a phytosanitary pest on the other crops and locations?
16. Table 1: Tetranychus “spp.”
17. Name headers of the results as the same as the methods if possible
18. Line 267 – figure S1 not S4
19. Lines 315-317, reorganize sentence in active voice to be more clear
20. Why did you only list strain Ma034 in the conclusion when in line 347 you also list Ma005 (however you list this one as variable - in reference to line 282)

Experimental design

Major points
1. Sampling - I suggest including a map of the area of the 5 sampling sites instead of the longitude and latitude in Table 1. You say they are far apart and would be nice to have some idea of ecological reservoirs near the sites
o It would help to have a section in methods describing the climate and ecology of the sampling sites since they have an effect on mite biology (although this is completely optional)
2. Sampling - Is there a reason you selected only two leaves from the bottom of the plants? Provide justification as to why only 2 leaves and why only on the bottom of the plants and not from other portions of the plant. How did you randomly select the fifteen plants? How large are the orchards and do they sufficiently represent the orchards?
3. Sampling – Do you have a clear image of both Tetranychus species (whole bodied and cleared) with the microscope? For better identification purposes in Figure 1. Do you have magnification levels and/or scale for size reference as well for all of the images in this figure?
o Figure 1a does show the aedeagus, however it would be better to see the whole mite specimen alongside the aedeagus pointed out or in combination with this image. And an image of the other species as well.
4. Colony – why were they raised on seedlings?
5. Entomopathogenic strains – you need more information on how the phylogenetic tree was created.
o What version of the UNITE database did you use? What version of Clustal Omega did you use?
o How did you get from the multi-sequence alignment to the tree?
6. Bioassays and infectivity – Consider separating the statistics out and place in the statistical analysis section

Minor points
7. Entomopathogenic – include what the sources of the ITS isolated were (table 4), you show in the table in the results that they came from different sources, but do not explain it in the methods
8. Bioassays – why was only T. merganser used in the bioassays and not the other species?
9. Bioassays – Figure 1b. include that these females were used in the mass screening for bioinfectivity in the caption
10. Line 139 - Qiagen not Quiagen – specify name of kit in addition to catalog number
11. Lines 167-168 – were the leaves cleaned in the same way but without the inoculum?

Validity of the findings

Major points
1. Presence – Show images of the difference between the two species under microscope if possible, and a picture of the yellow males compared to the red females.
2. Lines 264-268 on the ITS should come near the beginning of the results section (as they should in the methods section) and should continue with robertsii as the identifier and not the data still belonging to anisopliae.


Minor points
3. Line 214 – specific that you mean sampling dates
4. Line 229 – are you referring to the other fungal species strains or are you saying you think other strains of the H. thomposnii would work?
5. Figure 2 axis title “Mortality of female adult mites (%) needs to be consistent on all three charts
a. Indicate use of Tukey, error bars, and use of asterisks in figure caption
6. Separate out strain Bb016 from the other highest mortality strains and highlight it more
7. Data S1
a. LT50 not TL50
8. Table 2 – what is time measured in?
9. Table 4 – what do you mean by unidentified?
10. Line 276 – should not reference figure 1d unless you specify exactly which strain is photographed – since you are discussing multiple strains (at least which fungal species), or specify you mean they all look similar to that image

Additional comments

1. Clean experimental design and statistics, results look valid.
2. There is an overuse of commas in the introduction section specifically, which make the text harder to comprehend. Consider switching some of the sentences to more of an active voice and be more direct.
3. I like that you explain there are other naturally occurring entomopathogenic fungi that are specific to these families of mites, however they are productionally not a great prospect.
4. Good explanations of why some of the fungal isolates you used did not work. I would like to know a little information about the biological differences between the strains within a fungal species, though. Why does one strain work better than another – or this could be future research.

Reviewer 2 ·

Basic reporting

This manuscript presents sound bioassay data to show high potential of several M. anisopliae and B. bassiana strains against Tetranychus merganser, an emerging mite pest of papaya crops in Mexico. Up to 30 strains of different insect mycopathogens were screened in bioassays, resulting in recognition of two strains that enabled to cause high mortalities of mite female adults and eggs and hence are potential for the mite control. All experimental data were appropriately analyzed and presented, supporting the author's conclusion. The English writing is understandable but away from a standard for an international publication. Thus, the manuscript needs a major revision for improved clarity, conciseness and readability. I suggest the authors to consider professional editing service and an assistance from a colleague as a fluent English speaker.

Experimental design

The bioassays system used for assessment of fungal miticidal activity is technically acceptable. All data analyses are appropriate.

Validity of the findings

The experimental data support the authors' conclusion.

Additional comments

Some missed references (listed below) may help authors to expand their discussion on fungal infectivity to mite eggs and adults and also on biological control of spider mites by selected fungal strains.

1. Histopathological and molecular insights into the ovicidal activities of two entomopathogenic fungi against two-spotted spider mite. Journal of Invertebrate Pathology, 2014, 117: 73-78.
2. Effect of fungal infection on reproductive potential and survival time of Tetranychus urticae (Acari: Tetranychidae). Experimental and Applied Acarology, 2009, 48(3): 229-237.
3. Sprays of emulsifiable Beauveria bassiana formulation are ovicidal towards Tetranychus urticae (Acari: Tetranychidae) at various regimes of temperature and humidity. Experimental and Applied Acarology, 2008, 46: 247-257.
4. Time-concentration-mortality responses of carmine spider mite (Acari: Tetranychidae) females to three hypocrealean fungi as biocontrol agents. Biological Control, 2008, 46(3), 495-501.
5. Field trials of four formulations of Beauveria bassiana and Metarhizium anisoplae for control of cotton spider mites (Acari: Tetranychidae) in the Tarim Basin of China. Biological Control, 2008, 45(1): 48-55.
6. Field efficacy of application of Beauveria bassiana formulation and low rate pyridaben for sustainable control of citrus red mite Panonychus citri (Acari: Tetranychidae) in orchards. Biological Control, 2006, 39(3): 210-217.

---

## Round 0.2 · accepted · Accept

I have read the response and revised manuscript and I am satisfied that you have addressed the concerns of the reviewers. The manuscript is now acceptable for publication.